# A toolkit for planning and implementing acute febrile illness (AFI) surveillance

**Lilit Kazazian** *, **Rachel Silver**, **Carol Y. Rao**, **Michael Park**, **Chandler Ciuba**, **Madeline Farron**, **Olga L. Henao**

Global Epidemiology, Laboratory, and Surveillance Branch, Division of Global Health Protection, Global Health Center, Centers for Disease Control and Prevention, Atlanta, Georgia, United States of America

* pqr3@cdc.gov

**Data Availability Statement:** The authors confirm that all relevant data are within the paper and its Supporting Information files.

## Abstract

Acute febrile illness (AFI) is a broad clinical syndrome with a wide range of potential infectious etiologies. The lack of accessible, standardized approaches to conducting AFI etiologic investigations has contributed to significant global gaps in data on the epidemiology of AFI. Based on lessons learned from years of supporting AFI sentinel surveillance worldwide, the U.S. Centers for Disease Control and Prevention developed the toolkit for planning and implementing AFI surveillance, described here. This toolkit provides a comprehensive yet flexible framework to guide researchers, public health officials, and other implementers in developing a strategy to identify and/or monitor the potential causes of AFI. The toolkit comprises a cohesive set of planning aids and supporting materials, including an implementation framework, generic protocol, several generic forms (including screening, case report, specimen collection and testing, and informed consent and assent), and a generic data dictionary. These materials incorporate key elements intended to harmonize approaches for AFI surveillance, as well as setting-specific components and considerations for adaptation based on local surveillance objectives and limitations. Appropriate adaptation and implementation of this toolkit may generate data that expand the global AFI knowledge base, strengthen countries' surveillance and laboratory capacity, and enhance outbreak detection and response efforts.

## Introduction

Fever is a common sign of acute illnesses that may be caused by a diversity of pathogens with frequently non-specific clinical presentations [1, 2]. The etiologic agents of acute febrile illness (AFI) vary across geographic settings, seasons, and populations. However, knowledge of the context-specific causes of AFI remains limited, particularly in some low-resource settings without sufficient laboratory diagnostic capacity [2–5]. Global health researchers have increasingly called for more studies on the etiologies of AFI to inform clinical care, focus disease prevention efforts, and guide the allocation of public health resources [2–7].

A global scoping review of 190 AFI etiologic investigations published from 2005 to 2017 identified significant gaps in existing research approaches, including geographic disparities

**Funding:** The author(s) received no specific funding for this work.

**Competing interests:** The authors have declared that no competing interests exist.

and highly variable methodologies [3]. Siesel et al. updated this scoping review through 2019 and identified 51 new publications using the same methodology (**S1 File**) [8]. Despite the growing literature on AFI etiologies, some regions remain unrepresented. AFI case definitions and laboratory methods used varied widely across studies, hindering the aggregation and comparison of results across settings. In addition, most etiology investigations in the scoping review update tested for three or fewer pathogens, with single pathogen studies making up the greatest share. These literature gaps may be explained by local differences in research funding and prioritization, laboratory diagnostic capacity, and publication frequency [2, 4, 5]. Without expanded, multi-pathogen testing using appropriate and validated approaches across different settings, it is difficult to determine the scope of AFI etiologies, prioritize interventions, or consider co-infections.

Although some heterogeneity in AFI etiologic studies is expected, common methodological approaches may help improve data quality and comparability. Existing articles have proposed standard procedures for conducting AFI research, including study designs for estimating the incidence of causative pathogens [7] and a checklist for reporting study results [3]. A protocol for conducting a multi-site AFI etiologic study in Africa and Asia has also been published [9]. These approaches may mitigate some challenges associated with AFI *research*; however, no standards or recommendations exist for conducting routine AFI *surveillance* intended to collect critical data for public health decision-making. Robust epidemiological research is often resource-intensive and siloed into pathogen-specific systems, impeding the timely translation of data to practice. Routine surveillance using the syndromic envelope of AFI may generate more actionable information on circulating pathogens in low-resource settings. In addition to expanding the global AFI knowledge base, AFI surveillance can strengthen local surveillance and laboratory capacity, allowing for better detection and response to outbreaks [10–14].

The AFI surveillance toolkit, presented here, consists of an easy-to-adapt set of materials for conducting surveillance with the intent of identifying potential causes of fever, applicable to both routine surveillance and research activities conducted at healthcare and public health facilities. While strategies may vary widely depending on goals and context, the toolkit offers a basic approach for planning and implementing a surveillance system to generate useful information on the potential etiologies of AFI. Embedded throughout these materials are key elements intended to coordinate and harmonize activities and results across settings (e.g., regions, countries, provinces), in addition to setting-specific components that may be incorporated as applicable. Considerations for adapting the toolkit are also included to assist implementers in navigating common challenges and decision points.

## Methods

### Ethics statement

Individuals interviewed as part of the toolkit evaluation process provided verbal consent to participate during recorded interview sessions. This activity was reviewed by a CDC Human Subjects Representative and was conducted consistent with applicable federal law and CDC policy (project ID: 0900f3eb81f1d633). (See e.g., 45 C.F.R. part 46.102(l)(2), 21 C.F.R. part 56; 42 U.S.C. §241(d); 5 U.S.C. §552a; 44 U.S.C. §3501 et seq.)

### Development and evaluation process

The U.S. Centers for Disease Control and Prevention (CDC) has supported sentinel AFI surveillance in partner countries across Africa, Asia, Eastern Europe, and South and Central America [10, 12, 15]. In 2018, the CDC drafted a generic protocol as a starting point towards coordinating implementation. In July 2021, we established a multidisciplinary working group

to develop a comprehensive toolkit for AFI surveillance, expanding on the initial generic protocol. The working group comprised CDC subject matter experts (SMEs) in laboratory, epidemiology, surveillance, and information systems and regularly consulted with other colleagues involved in local AFI surveillance activities. **S2 File** lists members of the working group and SMEs and implementers who provided technical feedback. Based on the working group's review of existing country protocols and lessons learned from implementation, the group identified key components of an AFI surveillance protocol and considerations for adaptation based on local context. These key components and considerations were incorporated in a revised generic protocol and associated generic forms, as well as a newly developed implementation framework and protocol development checklist. To enhance data quality and comparability, the group also identified core, recommended, and setting-specific data elements, described in a generic data dictionary.

Between June 14th and July 11th, 2022, the authors conducted interviews as part of an iterative process to gather user input on the toolkit and assess the feasibility of implementation, objectives for use, and areas for improvement. LK interviewed nine individuals closely involved in AFI surveillance activities (pre-toolkit) supported by CDC country and regional offices in five geographic areas: Central America Region (CAR), Bangladesh, Liberia, Thailand, and Georgia. RS and CC both took detailed notes on each interview. A rapid analytic approach was used to synthesize key themes from consolidated interview notes [16, 17]. The analytic team (LK, RS, and CC) coded consolidated interview notes into eight domains drawn from the interview guide and refined based on interview content: surveillance planning challenges, surveillance implementation challenges, challenges using surveillance findings, key surveillance successes, useful toolkit materials, gaps in the toolkit, and ideas for improving toolkit. Codes were transferred into a matrix to synthesize key themes. Senior authors (OH, CR, and MP) reviewed and assisted with the interpretation of the findings. Feedback on the toolkit was positive overall, with participants finding particular value in the generic forms, data dictionary, and protocol development checklist. Interviews revealed the time-intensiveness of planning AFI surveillance to be a major challenge that could have been mitigated by the availability of a guiding framework. Specific proposed changes and additions to the toolkit were prioritized based on the level of effort and anticipated impact. Some changes were incorporated in this version of the toolkit (e.g., revised elements and considerations for adaptation), while the development of additional resources is ongoing (e.g., monitoring and evaluation framework, detailed laboratory guidance).

## Implementation methods

The AFI surveillance toolkit consists of interlinked, complementary materials for activity planning and implementation (**Fig 1**). Resources under the planning stage are intended to be referenced iteratively when defining the surveillance strategy. Once a surveillance protocol is established, the generic forms and data dictionary may be adapted and used during surveillance implementation.

As the foundation for the toolkit, the AFI surveillance implementation framework (**Fig 2**) outlines overarching questions and considerations to guide those seeking to implement AFI surveillance. The process begins with the identification of AFI surveillance objectives and continues through the establishment of specific methodologies, with ongoing assessment of the limitations that may affect the implementation of each component. Implementers may track their progression through the framework using the protocol development checklist of key decision points (**S3 File**). The generic protocol provides detailed considerations on each decision point and serves as a template for documenting surveillance procedures (**S4 File**).

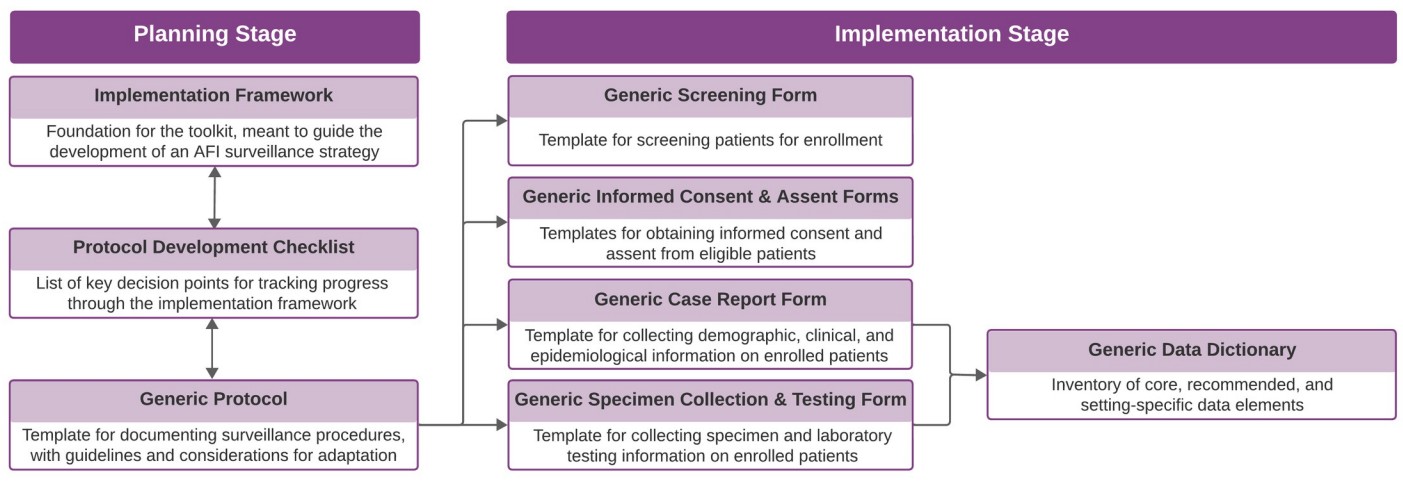

**Fig 1. AFI surveillance toolkit overview.**

The components of the AFI surveillance implementation framework are elaborated below, with links to the generic forms and tools corresponding to each component. Formulating a cohesive surveillance strategy based on this framework requires extensive discussion among diverse collaborators, including laboratorians, epidemiologists, health information system specialists, and public health officials.

## Why? Establishing the surveillance objective

Well-defined objectives are essential to guiding the development of an AFI surveillance strategy. The primary objective of most AFI surveillance centers around the identification and/or monitoring of priority AFI etiologies. Implementers may choose to incorporate elements of both approaches or transition from identifying to monitoring potential AFI etiologies over time as the knowledge base expands. Secondary objectives may include, but are not limited to, describing the epidemiology of AFI cases (e.g., age, sex, exposures, geography, seasonality), building local surveillance and laboratory capacity (e.g., data management systems, diagnostic testing, public health workforce), and informing public health priorities and specific aspects of clinical care (e.g., resource allocations, empiric treatment algorithms). Each of these objectives carries its challenges. Context-specific limitations, such as available resources, existing capacity, and government priorities, should all be carefully weighed when identifying objectives.

## Who? Identifying the target population

AFI surveillance objectives are closely tied to the population being investigated. Depending on surveillance priorities and enrollment feasibility, the population of interest may include or exclude specific groups based on demographics (e.g., children, agriculture workers, immunocompromised individuals), hospitalization status (inpatient versus outpatient), or symptom presentation. The enrollment of asymptomatic control cases, in addition to AFI patients, can allow the estimation of AFI risk factors and the etiologic fraction attributable to specific pathogens [18]. However, it may be resource-intensive and technically challenging to recruit control cases from a representative community-based population. The sampling framework represents an additional decision point: while enrolling all eligible individuals generally allows for the

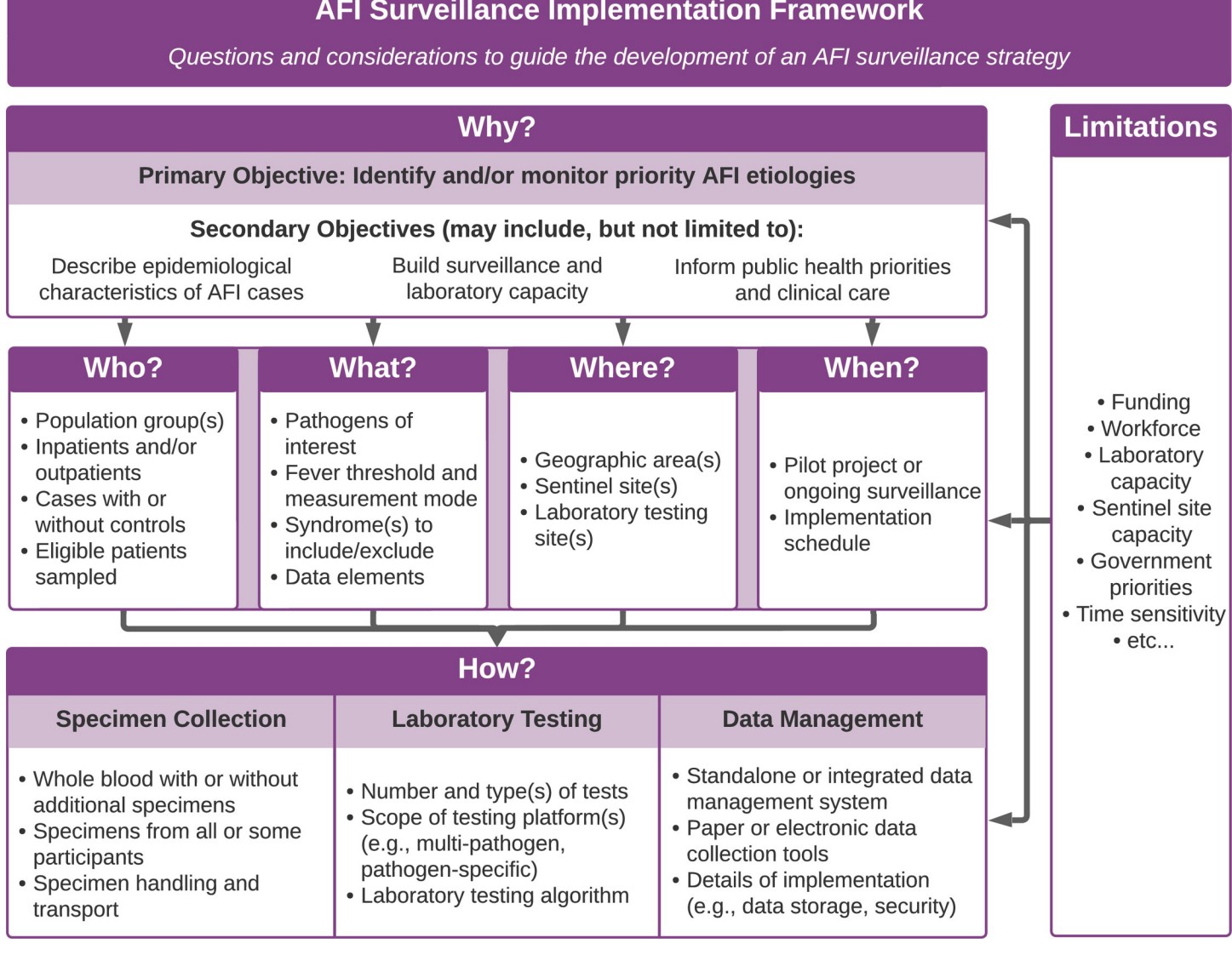

**Fig 2. AFI surveillance implementation framework.**

least biased estimates, a sampling scheme may be warranted depending on anticipated patient volume, available resources, and staffing constraints.

## What? Defining priority pathogens, cases, and data elements

The decision points in this portion of the framework are linked to nearly all other aspects of an AFI surveillance strategy. For instance, the choice of priority pathogens impacts the appropriate specimens to collect, laboratory tests to use, and population groups to recruit. In turn, existing knowledge gaps, government priorities, and resource and capacity constraints inform the number and selection of pathogens. The generic protocol offers guidelines for standardizing key components of an AFI case definition, including the fever threshold, temperature measurement mode, and exclusion criteria. These core components, in conjunction with the population of interest and sampling framework, form the basis of a generic screening form (S5 File).

The choice of demographic, clinical, and laboratory data elements to collect depends on AFI surveillance objectives and limitations, but some elements should preferably be maintained across settings. Core, recommended, and setting-specific data elements for AFI surveillance are defined in a generic data dictionary (**S6 File**). Each element in the data dictionary is associated with questions in the generic case report form (**S7 File**) and generic specimen collection and testing form (**S8 File**). Some questions and data elements may need to be refined after considering the decision points in the *How*? stage of the framework.

## Where? Selecting surveillance sites

The location and number of AFI surveillance sites determine the generalizability of surveillance findings given the geographic variation in population distribution and known AFI disease burden. Community health facilities, hospitals, and clinics commonly serve as sentinel sites for AFI patient enrollment. Sites with well-defined catchment areas may be preferred to permit the calculation of incidence rates. Other selection criteria for sentinel sites may include the presence of specific populations of interest, patient volume, known disease burden, degree of urbanization, government prioritization, opportunities to leverage existing platforms, laboratory infrastructure, specimen transport networks, and logistical concerns (e.g., distance, personnel, security). The location of potential external laboratory testing sites, and their distance from sentinel sites, should also be considered and further refined when establishing specimen collection and laboratory testing details.

## When? Defining the surveillance period

AFI surveillance may or may not be intended to be sustained over time. While a routine surveillance system may provide valuable ongoing information on circulating pathogens, a short-term research project may be more appropriate if the aim of the surveillance is limited to identifying common AFI etiologies. At least one year of surveillance data may be necessary to evaluate seasonal trends. The length of implementation and specifics of the implementation schedule, with defined stages for protocol development, training, enrollment, and data analysis, are likely to be influenced by practical considerations, such as funding availability and competing priorities.

## How? Operationalizing AFI surveillance

Once the basic elements of an AFI surveillance strategy begin to take shape, implementers may shift focus towards operationalizing this strategy. Decisions relating to specimen collection and laboratory testing procedures involve many technical and practical considerations and are often constrained by available resources and laboratory infrastructure. Selected specimens and laboratory tests should be validated for detecting target pathogens. Blood is nearly always used for detecting a wide range of febrile etiologies, but additional samples (e.g., stool, nasal swabs) may be recommended to determine the system infected, either among all patients or a subset (e.g., nasal swabs from those with respiratory symptoms). Laboratory tests often include a combination of polymerase chain reaction (PCR), rapid diagnostic tests, culture, or serological assays. Surveillance may begin with multiple-pathogen testing platforms [19–21], and transition to targeted pathogen-specific testing methods as priority pathogens are identified over time. The generic informed consent and assent forms should be adapted to reflect the selected specimens and laboratory tests to be conducted (**S9 File**).

Data management presents an additional set of decisions impacted by surveillance objectives, timelines, and available resources. Implementers may choose to maintain a standalone data management system or integrate AFI surveillance into existing national infectious disease

surveillance systems (e.g., DHIS2, LIMS), depending on feasibility and the intended scale of implementation. As an extension to the toolkit, a DHIS2 AFI surveillance module was developed by the University of Oslo in collaboration with CDC [22]. The module includes a data entry platform and a dashboard that allows users to monitor progress and results using real-time data visualizations. Real-time electronic data collection enhances data quality and timeliness but may not be feasible due to higher start-up costs or connectivity issues. Other data management matters, including data storage and security measures, should be tailored to the overall surveillance workflow and in line with local standards. Training for both laboratory and surveillance staff is a critical piece of operationalizing an AFI surveillance strategy that will need to be accounted for in budgeting decisions and timelines for implementation.

## Discussion

Given the well-documented global gaps in knowledge on the causes of fever, there is a need for innovative approaches to increase our understanding of AFI etiologies across diverse settings. To date, the absence of a comprehensive framework for AFI surveillance has limited the utility and comparability of data on AFI [2–5]. Our proposed AFI surveillance framework, generic protocol, and associated tools aim to guide public health officials, implementers, and researchers through the decision-making required to identify or monitor the potential causes of AFI. We propose templates and supporting materials based on lessons learned in practice in various countries. These resources address the need for some methodological standardization while acknowledging differences in surveillance objectives and limitations.

The AFI surveillance toolkit is the first resource of its kind to provide strategic guidance on establishing an AFI surveillance system–balancing technical considerations with the importance of leveraging and strengthening existing surveillance and laboratory capacity. Relative to traditional research protocols [9, 23], the proposed generic protocol is more flexible to adapt to local context but includes detailed considerations to simplify implementation in low-resource settings. Implementers may choose to integrate AFI surveillance into existing surveillance and laboratory systems, resulting in sustainable improvements in the public health workforce and infrastructure needed to improve the detection of and response to global health security threats [24, 25]. The proposed core case definition and data elements establish a baseline to compare AFI surveillance findings across settings, enabling joint investigations of infectious diseases with a febrile component, including those caused by emerging pathogens. CDC-supported global AFI surveillance activities have been successfully adapted to identify and respond to several outbreaks of emerging infectious diseases, including dengue [11], Zika [13, 26], and COVID-19 [14, 15].

The inherent flexibility of the AFI surveillance toolkit imposes several limitations on its potential uses. First, the toolkit does not include a surveillance protocol ready for immediate implementation. While the toolkit eases the burden of developing these materials, significant time and effort are needed to adapt the generic protocol and associated tools to the local context. Second, adapted AFI surveillance protocols are likely to vary widely across settings unless there is early and intentional coordination between implementers. Differences in patient enrollment, temporality, specimen collection, and laboratory testing may all affect the comparability and possibility of pooling of findings, even while core data elements enable data harmonization. Third, the toolkit primarily guides users in approaches to identify or monitor potential AFI etiologies, without necessarily attributing AFI cases to specific pathogens. Investigating whether detected pathogens are responsible for observed AFI clinical manifestations would require careful adaptations, such as enrolling asymptomatic controls or assessing clinical data in conjunction with laboratory test results [6, 27]. Fourth, the ability to extrapolate

findings to a larger population–for example, to calculate incidence or mortality rates–requires knowledge on the composition of the sample relative to the reference population, including care-seeking behaviors [7]. If implementers intend to make population-level inferences, community-based or "hybrid" surveillance may be more appropriate than sentinel surveillance approaches [28, 29].

The AFI surveillance framework, generic protocol, and associated tools will continue to be refined based on lessons learned through adaptation and implementation by CDC partner countries. These materials can be further evaluated by multisectoral groups interested in or working on AFI surveillance. Future iterations may need to consider how advances in laboratory technologies, such as next-generation sequencing, may impact AFI surveillance [30]. Ultimately, the toolkit can serve as a foundation for the development of coordinated global AFI surveillance activities. The expansion of coordinated implementation of high-quality AFI surveillance will improve the knowledge base of the causes of febrile illness by public health practitioners and policymakers worldwide, enabling more targeted prevention and response efforts.

## Supporting information

**S1 File. Publications identified from global AFI scoping review update, 2018–2019.** The scoping review update followed methods previously described in: Rhee C, Kharod GA, Schaad N, Furukawa NW, Vora NM, Blaney DD, et al. Global knowledge gaps in acute febrile illness etiologic investigations: A scoping review. PLOS Neglected Tropical Diseases. 2019;13(11): e0007792.
(XLSX)

**S2 File. Members of working group for development of AFI surveillance toolkit and technical reviewers of toolkit.**
(DOCX)

**S3 File. AFI surveillance protocol development checklist.**
(DOCX)

**S4 File. AFI surveillance generic protocol.**
(DOCX)

**S5 File. AFI surveillance generic screening form.**
(DOCX)

**S6 File. AFI surveillance generic data dictionary.**
(XLSX)

**S7 File. AFI surveillance generic case report form.**
(DOCX)

**S8 File. AFI surveillance generic specimen collection and testing form.**
(DOCX)

**S9 File. AFI surveillance generic informed consent and assent forms.**
(DOCX)

## Acknowledgments

We are grateful to the interview participants who contributed their time and feedback towards the continued improvement of this toolkit.

**Disclaimer:** The findings and conclusions in this report are those of the author(s) and do not necessarily represent the official position of the U.S. Centers for Disease Control and Prevention.

## Author Contributions

**Conceptualization:** Lilit Kazazian, Rachel Silver, Carol Y. Rao, Michael Park, Madeline Farron, Olga L. Henao.

**Methodology:** Lilit Kazazian, Rachel Silver, Carol Y. Rao, Chandler Ciuba.

**Project administration:** Lilit Kazazian, Rachel Silver, Carol Y. Rao.

**Supervision:** Olga L. Henao.

**Validation:** Lilit Kazazian, Rachel Silver, Chandler Ciuba.

**Writing – original draft:** Lilit Kazazian.

**Writing – review & editing:** Lilit Kazazian, Rachel Silver, Carol Y. Rao, Michael Park, Chandler Ciuba, Madeline Farron, Olga L. Henao.

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
