## [Decision Letter · Decision Letter 0]

14 Dec 2023

PGPH-D-23-01745

A toolkit for planning and implementing acute febrile illness (AFI) surveillance

Dear Dr. Kazazian,

Thank you for submitting your manuscript to PLOS Global Public Health. After careful consideration, we feel that it has merit but does not fully meet PLOS Global Public Health’s publication criteria as it currently stands. Therefore, we invite you to submit a revised version of the manuscript that addresses the points raised during the review process.

We look forward to receiving your revised manuscript.

Kind regards,

Raquel Muñiz-Salazar, Ph.D.

Academic Editor

Journal Requirements:

1. We ask that a manuscript source file is provided at Revision. Please upload your manuscript file as a .doc, .docx, .rtf or .tex.

Additional Editor Comments (if provided):

The primary recommendation from the reviewers is to provide empirical evidence demonstrating the effectiveness and applicability of the toolkit in real-world scenarios. This could be achieved by conducting a pilot study or presenting field data from a target population where the toolkit has been applied. Such data would not only validate the toolkit's practical utility but also strengthen the manuscript's argument by showing tangible results. Besides, the authors should include a comprehensive visual representation of the toolkit's components and their interrelationships. This would simplify understanding and enhance the manuscript's clarity.

Reviewers' comments:

Reviewer's Responses to Questions

**Comments to the Author**

1. Does this manuscript meet PLOS Global Public Health’s publication criteria? Is the manuscript technically sound, and do the data support the conclusions? The manuscript must describe methodologically and ethically rigorous research with conclusions that are appropriately drawn based on the data presented.

Reviewer #1: Yes

Reviewer #2: Yes

2. Has the statistical analysis been performed appropriately and rigorously?

Reviewer #1: N/A

Reviewer #2: Yes

3. Have the authors made all data underlying the findings in their manuscript fully available (please refer to the Data Availability Statement at the start of the manuscript PDF file)?

Reviewer #1: No

Reviewer #2: Yes

4. Is the manuscript presented in an intelligible fashion and written in standard English?

Reviewer #1: Yes

Reviewer #2: Yes

5. Review Comments to the Author

Reviewer #1: In the manuscript "A toolkit for planning and implementing acute febrile illness (AFI) surveillance" by Lilit Kazazian and colleagues, submitted to PLOS Global Public Health, the authors developed the toolkit for planning and implementing Acute febrile illness (AFI) surveillance. The toolkit provides a comprehensive framework to guide researchers and public health officials, to identify and/or monitor the potential causes of AFI. The toolkit includes a set of planning aids and supporting materials, an implementation framework, generic protocol, several generic forms (including screening, case report, specimen collection and testing, and informed consent and assent), and a generic data dictionary. The authors claim these materials incorporate key elements intended to harmonize approaches for AFI surveillance, as well as setting-specific components and considerations for adaptation based on local surveillance objectives and limitations. Finally, the authors suggest that the appropriate adaptation and implementation of the toolkit may generate data that expand the global AFI knowledge base, strengthen countries’ surveillance and laboratory capacity, and enhance outbreak detection and response efforts.

The study was well conducted, the documents generated are comprehensive and carefully produced, and the manuscript is clear and well presented. The value of the documents presented here are of great relevance and should be adopted and implemented in relevant settings.

Concerns and comments:

1) The toolkit was developed with the input of many experts. However, was the toolkit ever tested in a pitot study or target population? Is there some field data that could be presented here as an example of the application of the toolkit?

2) When applying the toolkit in a proper setting, large amount of data will be generated. Despite the suggestion of using DHIS2 or LIMS, is there a program developed or being developed specifically to receive data coming from the toolkit forms and analyze it fast and efficiently?

3) For the sake of simplicity, please include a workflow diagram describing all the documents of the toolkit, including the set of planning aids and supporting materials, the implementation framework, generic protocol, generic forms (including screening, case report, specimen collection and testing, and informed consent and assent), and a generic data dictionary.

Reviewer #2: The toolkit provides a flexible framework for researchers and public health officials to plan and implement AFI surveillance. It includes an implementation framework, a generic protocol, forms, and a data dictionary, aiming to harmonize approaches and adapt to local needs. It emphasizes the need for methodological standardization while accommodating different approaches and goals.

I think this will be very helpful and I can imagine recommending this to students/colleagues.

I did when initially opening the paper think that it was going to be a very different paper with an algorithm and workflow for particular testing and contexts, including more on microbiology decisions. So it may help to clarify this more in the title and abstract. I thought halfway through the abstract this might be a mini lab approach to fever.

I think this can be accepted as it is now and have said this separately as well, but it would be improved with a refinement of the who/what/when/where/why sections. Most of the explanations are just listing of information. Giving perhaps a pull out example of particular cases and how it is used would help or going into more depth, succinctly, of how the context and population were determined to affect the use of the tool.

6. PLOS authors have the option to publish the peer review history of their article (what does this mean?). If published, this will include your full peer review and any attached files.

**Do you want your identity to be public for this peer review?** For information about this choice, including consent withdrawal, please see our Privacy Policy.

Reviewer #1: No

Reviewer #2: No

---

## [Decision Letter · Decision Letter 1]

26 Mar 2024

A toolkit for planning and implementing acute febrile illness (AFI) surveillance

PGPH-D-23-01745R1

Dear Mrs. Kazazian,

We are pleased to inform you that your manuscript 'A toolkit for planning and implementing acute febrile illness (AFI) surveillance' has been provisionally accepted for publication in PLOS Global Public Health.

Best regards,

Raquel Muñiz-Salazar, Ph.D.

Academic Editor

Thank you for submitting your manuscript and for your patience during the review process. We have received feedback from our reviewers, and after careful consideration, I would like to share their insights and our decision on the next steps for your manuscript.

Reviewer 01 has indicated satisfaction with the revisions you have made, noting that all comments have been addressed with no further concerns. This is indeed a positive outcome, reflecting your commitment to enhancing the quality of your work.

However, Reviewer 02 has provided some constructive feedback that we believe could further strengthen your manuscript.

I have decided to accept the revised version of the manuscript for publication.

Reviewer Comments (if any, and for reference):

Reviewer's Responses to Questions

**Comments to the Author**

1. If the authors have adequately addressed your comments raised in a previous round of review and you feel that this manuscript is now acceptable for publication, you may indicate that here to bypass the “Comments to the Author” section, enter your conflict of interest statement in the “Confidential to Editor” section, and submit your "Accept" recommendation.

Reviewer #1: All comments have been addressed

Reviewer #2: All comments have been addressed

2. Does this manuscript meet PLOS Global Public Health’s publication criteria? Is the manuscript technically sound, and do the data support the conclusions? The manuscript must describe methodologically and ethically rigorous research with conclusions that are appropriately drawn based on the data presented.

Reviewer #1: Yes

Reviewer #2: Yes

3. Has the statistical analysis been performed appropriately and rigorously?

Reviewer #1: N/A

Reviewer #2: N/A

4. Have the authors made all data underlying the findings in their manuscript fully available (please refer to the Data Availability Statement at the start of the manuscript PDF file)?

Reviewer #1: No

Reviewer #2: Yes

5. Is the manuscript presented in an intelligible fashion and written in standard English?

Reviewer #1: Yes

Reviewer #2: Yes

6. Review Comments to the Author

Reviewer #1: I have no concerns with this manuscript.

Reviewer #2: The article would have been reframed to make expectations clearer. Readers will expect more information on piloting the study and on what toolkit can encompass

7. PLOS authors have the option to publish the peer review history of their article (what does this mean?). If published, this will include your full peer review and any attached files.

**Do you want your identity to be public for this peer review?** For information about this choice, including consent withdrawal, please see our Privacy Policy.

Reviewer #1: No

Reviewer #2: No
